# The Coming Age of Antisense Oligos for the Treatment of Hepatic Ischemia/Reperfusion (IRI) and Other Liver Disorders: Role of Oxidative Stress and Potential Antioxidant Effect

**DOI:** 10.3390/antiox13060678

**Published:** 2024-05-31

**Authors:** Siyuan Yao, Aanchal Kasargod, Richard Chiu, Taylor R. Torgerson, Jerzy W. Kupiec-Weglinski, Kenneth J. Dery

**Affiliations:** The Dumont-UCLA Transplantation Center, Department of Surgery, Division of Liver and Pancreas Transplantation, David Geffen School of Medicine at UCLA, Los Angeles, CA 90095, USA

**Keywords:** antisense oligonucleotides, CEACAM1, hepatocytes, inflammation, ischemia-reperfusion injury, Kupffer cells, organ transplantation, oxidative stress

## Abstract

Imbalances in the redox state of the liver arise during metabolic processes, inflammatory injuries, and proliferative liver disorders. Acute exposure to intracellular reactive oxygen species (ROS) results from high levels of oxidative stress (OxS) that occur in response to hepatic ischemia/reperfusion injury (IRI) and metabolic diseases of the liver. Antisense oligonucleotides (ASOs) are an emerging class of gene expression modulators that target RNA molecules by Watson–Crick binding specificity, leading to RNA degradation, splicing modulation, and/or translation interference. Here, we review ASO inhibitor/activator strategies to modulate transcription and translation that control the expression of enzymes, transcription factors, and intracellular sensors of DNA damage. Several small-interfering RNA (siRNA) drugs with N-acetyl galactosamine moieties for the liver have recently been approved. Preclinical studies using short-activating RNAs (saRNAs), phosphorodiamidate morpholino oligomers (PMOs), and locked nucleic acids (LNAs) are at the forefront of proof-in-concept therapeutics. Future research targeting intracellular OxS-related pathways in the liver may help realize the promise of precision medicine, revolutionizing the customary approach to caring for and treating individuals afflicted with liver-specific conditions.

## 1. Introduction

A fundamental aspect of cellular homeostasis involves the intricate balance between cellular oxidative processes and antioxidant defense mechanisms. Though all living organisms produce reactive oxygen species (ROS) as a byproduct of normal cellular metabolic pathways, at high concentrations, the balance shifts towards posing a threat to the integrity of cell structures such as carbohydrates, nucleic acids, lipids, and proteins, thereby altering their functions [1]. Serving as essential signaling molecules and, in some cases, acting as the principal toxic agents through which inflammatory cells eliminate their targets, such as bacteria, the scavenging capacity of ROS acts by promoting mitochondrial dysfunction, leading to oncotic necrosis [2].

In the liver, an organ crucial for metabolic homeostasis and detoxification, oxidative stress (OxS) contributes to the progression of liver pathogenesis by promoting injury to parenchymal hepatocytes, the principal cells responsible for metabolic activities such as detoxification, protein synthesis, and bile production. OxS that injures liver non-parenchymal cells, such as hepatic stellate cells, can lead to excessive extracellular matrix deposition and cirrhosis [3]. The surge in ROS and reactive nitrogen species (RNS) can be considered the initial inflammatory step in the pathogenesis of many types of liver disease, including non-alcoholic fatty liver disease (NAFLD), alcoholic liver disease (ALD), viral hepatitis B and C, and liver fibrosis and cirrhosis [4]. As one of the central metabolic hubs, the liver is particularly accessible for the development of novel therapeutics [5]. The liver regulates energy and lipid metabolism and, as part of the body’s central metabolic organ, it has potent immunological functions. In the past decade, small antisense oligonucleotide (ASO) molecules have proved to be an effective approach for targeting genes linked to human diseases. The rationale of targeting specific mRNAs to modulate protein expression has broadened the set of therapeutic targets that can be linked to mitigating human diseases.

Small-molecule therapeutics encompass various nucleic-acid-derived compounds that interact with specific nucleotide mRNA sequences within cells. Small-molecule drugs differ in their origin, chemical structure, and mode of action. Some are naturally occurring, e.g., microRNAs (miRNAs) and ribozymes, whereas many others, such as short activating RNAs (saRNAs), locked nucleic acids (LNAs), Gapmers, and Phosphorodiamidate Morpholino Oligomers (PMOs), are synthetically designed for optimized potency, selectivity, and pharmacokinetic properties [6]. A summary of the different types of well-characterized ASOs is presented in Figure 1. Generally, ASOs are defined as chemically synthesized oligonucleotides, 12–30 long nucleotides, designed to bind to RNA by Watson–Crick base pairing rules [7]. By using nucleotide hybridization as the primary means to achieve specificity, stability, and efficiency of the resulting duplex structure [8], ASOs can modulate gene expression through various mechanisms, including inhibition of translation, induction of mRNA degradation via RNase H-mediated cleavage, or alteration of mRNA splicing patterns (Figure 2).

The field of antisense technology has undergone different phases of development, from initial enthusiasm to a period of disillusionment; the numerous clinical trials currently in progress provide grounds for renewed optimism. For the purpose of this review, where possible, primary publications will focus on the intricate interplay between cellular OxS and ROS, with a special focus on how ASOs have been used to decorticate OxS-mediated liver-related pathology and disease. We review recent ASO studies that offer deeper insight into biological mechanisms, focusing on next-generation molecular targets. Additionally, we examine the status of small-molecule clinical trials (Phase 1 and 2) that harness basic mechanistic insights into next-generation treatments in chronic liver diseases and associated fibrotic pathogenesis (Table 1).

## 2. OxS in Liver Ischemia-Reperfusion Injury

Our research studies have focused primarily on the role of OxS in ischemia-reperfusion injury (IRI). However, the contribution of ROS to the development of non-alcoholic steatohepatitis (NASH), NAFLD, and alcoholic steatohepatitis (ASH) is well-appreciated as well [24,25]. IRI complications inevitably arise during surgical hepatectomy procedures when, during organ transplantation, grafts are first transported and preserved under cold storage conditions. The decrease in oxygen supply to liver tissue causes ischemic stress and leads to an imbalance between ROS production and antioxidant defense mechanisms [26].

During hepatic IRI, enzymes like xanthine oxidase and NADPH oxidase in the mitochondria initiate the production of ROS [27]. An inflammatory cascade, termed nitro-OxS, follows with inducible nitric oxide synthase (iNOS), leading to nitric oxide (NO) build-up, and in the presence of O_2_^•^, highly reactive peroxynitrite (ONOO-) forms [28]. In many cases of pathological IRI, peroxynitrite can be further converted to even more reactive radicals, such as nitrogen dioxide (^•^NO_2_) [29]. During this phase of IRI, specialized parenchymal macrophage cells, such as Kupffer cells (KCs), generate ROS that exacerbate DNA and organelle damage at the hepatocyte–liver sinusoidal endothelial cell (LSEC) interface. This results in the production and local secretion of redox-sensitive damage-associated molecular patterns (DAMPs), such as high mobility group box 1 (HMGB1), into the extracellular space. The resulting damage to the liver includes significant microcirculatory dysfunction, impaired vasodilation, activation of endothelial cells, increased vascular resistance and inflammation in the liver, infiltration of leukocytes, OxS, and eventual cell death (Figure 3). Later, as oxygen is reintroduced during surgical procedures, reperfusion of blood flow brings an influx of ROS and DAMPs that activate a classic positive amplification feedback loop, initiating an immune cascade that causes further damage to cells and inflammation in the affected hepatic tissues [30]. Recent research suggests that liver transplants experiencing severe IRI show worse outcomes, have a higher incidence of early allograft dysfunction (EAD), and have poorer overall survival rates [31,32,33,34]. EAD is a clinical syndrome characterized by impaired function of the transplanted liver in the immediate postoperative period, typically within the first week after transplantation [31]. Therefore, targeting antioxidant pathways presents a promising treatment opportunity.

## 3. OxS in Other Liver Diseases

OxS is a key pathogenic factor in the progression of many liver diseases, such as NAFLD, an increasingly common worldwide condition that is asymptomatic in its early stages [36,37]. NAFLD is actually a spectrum of liver conditions ranging from simple steatosis (fatty liver) to NASH, which involves inflammation and liver cell damage and can progress to fibrosis, cirrhosis, and even liver cancer in severe cases [38]. Studies show that patients with NAFLD exhibit higher levels of OxS and lipid peroxidation products in their serum/plasma blood fluid [39]. ROS typically found in NAFLD patients derives from mitochondrial-generated superoxide anions (O_2_^•−^), which are byproducts of oxidative phosphorylation, and peroxisomes, which function by breaking down long-chain fatty acids in a process called beta-oxidation. As chronic liver injury progresses, protective antioxidant defense mechanisms fail to overcome the induction of OxS-sensitive transcription factors, such as NF-kB, Egr-1, and AP-1, that ultimately lead to hepatocyte cell death [40,41].

Emerging evidence suggests a role for cellular networks that crosstalk with our immune cells and gut microbiota in facilitating chronic liver disease progression. For example, a recent study on NAFLD and cirrhosis described a subpopulation of human resident liver myeloid cells (LM) that were protective against obesity-associated OxS development [42]. LMs were shown to upregulate Peroxiredoxin 2 (PRDX2), a biological catalyst that reduces hydrogen peroxide, organic hydroperoxides, and peroxynitrite, essential for detoxifying harmful compounds. Though functional validation was demonstrated using human 2D and 3D cultures, it remains to be seen whether LMs alleviate the OxS load once metabolic diseases linked to obesity have been initiated.

The disturbances to the gut microbiota, such as dysbiosis, also play a role in OxS [43]. In one recent study, Song et al. explored the relationship between gut microbiota and valproate (VPA)-induced hepatic steatosis, which is associated with OxS. VPA is used to treat neurological disorders such as epilepsy and bipolar disorder and acts by increasing the levels of gamma-aminobutyric acid (GABA) neurotransmitters in the brain [44,45]. It helps to calm overactive electrical signaling that can lead to seizure disturbances. Researchers found that VPA treatment led to increased CYP2E1, a pro-oxidant gene, as well as the downregulation of catalase (CAT), glutathione S-transferase (GST), superoxide dismutase (SOD), and heme oxygenase 1 (HO-1), all of which are part of the Nuclear factor erythroid 2-related factor 2 (Nrf2) transcriptional pathway. Conversely, probiotic administration ameliorated conditions, reducing OxS in the presence of VPA by decreasing CYP2E1 levels and activating the Nrf2 pathway. Analyses of bacteria from the VPA group that induced hepatic liver steatosis showed higher levels of *Actinobacteriota*, *Acidobacteriota*, and *Gemmatimonadota*, whereas probiotic treatment significantly reversed the changes.

Other studies have explored the potential of using antioxidant compounds to mitigate OxS-related liver damage. In a recent report, the antioxidant compound verbenalin was studied for its role in regulating mitochondrial dysfunction in a model of ASH pathogenesis [24]. Verbenalin works by modulating various signaling pathways involved in inflammation and cellular damage, scavenging free radicals, and inhibiting OxS, thereby protecting cells and tissues from damage caused by ROS. Dong et al. showed that verbenalin alleviates ROS and lipid peroxidation levels via MDMX (Murine double minute X)/PPARα (Peroxisome proliferator-activated receptor alpha)-mediated ferroptosis, a type of regulated cell death characterized by iron-dependent lipid peroxidation [46]. Similarly, Su et al. investigated the activation of Stimulator of interferon genes (STING) by OxS-mediated ferroptosis in liver fibrosis and carcinogenesis in the absence of transforming growth factor-beta-activated kinase 1 (TAK1) [47]. TAK1 deletion led to increased hepatocyte ferroptosis, marked by elevated ROS, lipid peroxidation, MDA, 4-HNE, and decreased the glutathione (GSH) to oxidized glutathione (GSSG) ratio by activating the Nrf2 pathway. Moreover, in TAK1-deficient tissues, oxidative DNA damage from hepatocyte ferroptosis increased macrophage cGAS-STING activation and intrahepatic inflammation. This was alleviated by Fer-1 and anti-8-OHG (8-hydroxyguanosine) antibody treatment, respective inhibitors for ferroptosis and oxidative DNA damage.

Other compounds that reduce OxS in the liver include Metformin (Met) and vitamins A, C, and E. As an allosteric regulator of mitochondrial glycerophosphate dehydrogenase (mGPD), Met alters the balance of NADH and NAD+ within the cell, leading to reduced conversion of lactate and glycerol to glucose, which in turn leads to decreased hepatic gluconeogenesis [48]. Studies have shown that Met improves the sensitivity of peripheral tissues to insulin, including the liver, making it a first-line drug in type 2 diabetes mellitus therapy [49]. A recent study investigated whether adipose mesenchymal stem cell-derived exosomes (ADSCs-Exo), functioning as a vehicle to deliver Met, would provide a mitochondrial protective role in the treatment of hepatic IRI [50]. The study showed that the application of ADSCs-Exo in vivo was effective in inhibiting mitochondrial fission-related protein expression through the AMPK (AMP-activated protein kinase) and SIRT1 (Sirtuin 1) signaling pathway. Moreover, naturally occurring polyphenolic compounds found in various plants, including grapes, berries, and peanuts, have also been shown to reduce OxS in the liver. For example, Resveratrol and Quercetin were shown to be combinatorially effective at reducing fatty liver in a recent study investigating metabolic syndrome (MS) in rats [51]. The mechanism was attributed to the over-expression of the master factor Nrf2, which in turn led to the increase in antioxidant enzymes (catalase, peroxidases, glutathione-S-transferase, glutathione reductase) and GSH (reduced glutathione) recycling. Taken together, these studies demonstrate the need for new therapeutic antioxidant modalities to alleviate cellular damage and improve liver health. As ASOs have shown promise in the treatment of a range of conditions such as neurological disorders, cardiovascular diseases, and inflammatory disorders such as asthma and inflammatory bowel disease (IBD) [52,53,54,55], their potential use in organ transplantation may offer the precision medicinal intervention needed to tailor individual genetic profiles to reduce the incidence of EAD and IRI.

## 4. The Duality of ASOs as a Therapeutic Intervention

To achieve their intended pharmacological effects, ASOs must traverse biological membranes within the cytoplasm or nucleus to find their target, such as preRNA, mRNA, non-coding RNA, and toxic nuclear-localized RNA [56]. Once they bind to the target RNA, ASOs or the antisense strand of the RNA duplex can impact the RNA’s metabolism by altering the mRNA stability and turnover rates [57,58]. ASOs have been designed in an assortment of chemical classes, with the majority featuring phosphorothioate backbones coupled with one or more 2′-ribose sugar modifications (such as 2′-O-methyl (2′-methoxyethyl) (2′-MOE), 2′-O-(2-methoxyethyl) cytidine (cEt), LNA, and 2′-O-methyl (2′-OMe), or they may possess sugar-phosphate modifications like morpholino and peptide nucleic acids (PNA) [56]. Some ASOs exhibit dual functionality, e.g., LNAs, gapmers, and mixmers, capable of silencing and activating gene expression depending on their design and target sequence. This intriguing feature underscores the versatility of ASOs as potent tools for finely tuning gene regulation in various therapeutic and research applications [59].

The versatility of ASOs that makes them good candidates for therapeutic application stems from the high degree of sequence specificity that leads to the degradation, modulation, or manipulation by alternative splicing of target RNAs [60]. Their combined sensitivity and specificity to both enhance and reduce protein expression sets them apart from siRNAs or microRNAs, which are generally restricted to silencing targeted expression [61]. Another advantage of ASOs is their relatively straightforward design and production, so the reduced time between conceptualization and clinical use offers the possibility of rapid development of patient-customized treatments. In one promising example, the development of Milasen, a splice-modulating antisense oligonucleotide drug, was designed and tested 1 year after first contact with a single six-year-old patient with Batten disease, a rare, fatal, inherited disorder of the nervous system [62]. ASOs solve the intractable challenge of targeting RNA transcripts of genes that are considered “undruggable” by conventional approaches, offering a clinician an armamentarium that treats a wider range of genetic disorders [63]. Other significant advantages include long-lasting effects, minimal immune responses, and applications across various tissues. This was demonstrated recently in a study of dystrophia myotonica type 1 (DM1), a multi-systemic genetic disorder characterized by progressive muscle wasting and weakness [64]. ASOs (IONIS 486178 ASO) delivered to the central nervous system led to a 30–50% reduction in human Dystrophia Myotonica Protein Kinase (DMPK) mRNA 12 weeks after injection in mice.

The cellular uptake of ASOs as a therapy is not without its own challenges and limitations. First, the hydrophobic nature of the phospholipid membrane may hinder the ability of ASOs to target RNA molecules. A strategy using chemical ligation of palmitate, tocopherol, and cholesterol to plasma proteins such as albumin and lipoproteins has been shown to be effective in targeting extra-hepatic tissues in mice [65]. Similarly, endosomal entrapment may inadvertently lead to nuclease digestion susceptibility. To overcome this, studies show that OECs (oligonucleotide enhancing compounds), such as sodium butyrate, are effective at perturbating multivesicular bodies (MVB), a type of endosome involved in the sorting and trafficking of cellular components [66,67]. Sodium butyrate is a histone deacetylase (HDAC) inhibitor that can enhance the efficacy of ASOs by improving their uptake and activity within cells [68,69]. Unmethylated CpG motifs may elicit immune responses that lead to adverse effects or decreased efficacy. These considerations may also lead to off-target effects by ASOs that inadvertently interact with unintended RNA targets that, lead to undesirable biological consequences, and require time-consuming optimization in the clinical setting. Finally, the challenge of choice of administration route (e.g., systemic injection, local injection, or oral administration) as well dose optimization may be needed to reduce the chance of hepato- and renal toxicity, with each route presenting unique challenges and considerations.

## 5. Modulation of Signaling Pathways Involved in Liver IRI by ASO

The effectiveness of ASOs was initially showcased in chick embryos, where synthetic oligonucleotides hindered the translation of Rous sarcoma viral RNA, inhibiting viral replication [70]. Since then, ASOs have emerged as a promising avenue for mitigating IRI by targeting specific genes and modulating key signaling pathways involved in the injury cascade. Studies have focused on the role of ASOs in parenchymal cells, primarily hepatocytes, that constitute the bulk of the liver tissue and resident hepatic KC macrophages. However, other liver cell types (e.g., hepatic stellate cells (HSCs), LSECs, and bile duct epithelial cells (cholangiocytes)) have been investigated recently [71,72,73].

### 5.1. ASOs Targeting Liver Hepatocytes

Since apoptosis and necrosis are the primary manifestations of hepatic IRI, it is essential to understand the contributing factors that induce hepatocyte cell death [74]. Our group approached this question by investigating the mitogen-activated protein kinase (MAPK) signaling ASK1 (apoptosis signal-regulating kinase 1) and p-p38 cell-death cascade [75]. Importantly, ROS directly activates ASK1 by inducing the dissociation of thioredoxin from ASK1, allowing ASK1 to become phosphorylated and activated [76]. Phosphorylated p38 (p-p38) MAPK subsequently initiates a signaling cascade that activates various downstream targets, including transcription factors and other kinases, ultimately resulting in cellular responses such as inflammation, apoptosis, or cell proliferation [77]. We showed that hepatocyte CEACAM1 (CD66a), the gene that encodes a transmembrane protein called carcinoembryonic antigen-related cell adhesion molecule-1, deficiency enhanced the p-p38 increase and HMGB1 DAMP translocation, consistent with OxS stress-dependent injury markers seen in the liver IRI cascade. By co-culturing cells with an ASK1-siRNA, Nakamura et al. showed that cold stress triggered an increase in ASK1/p-p38 MAPK expression, whereas CEACAM1 signaling exerted potent hepatoprotection by suppressing the ASK1/p38 MAPK signaling axis [75]. Another group recently investigated the role of MEK/ERK activity associated with the upregulation of pro-apoptotic factor Bax and the downregulation of anti-apoptotic factor Bcl-2. Studies have shown that ERK activation triggers apoptosis when observed in hepatic IRI. Consequently, inhibiting the MEK/ERK pathway offers a protective mechanism against liver damage by mitigating inflammatory responses and apoptosis. Another study showed that when the liver is subjected to IRI stress, the expression of CCN1 (Cellular communication network factor 1) is strongly induced, which leads to the MEK/ERK pathway activation in hepatocytes. By applying a CCN1-siRNA, Liu et al. showed that a CCN1 knockdown reduced liver enzyme levels, myeloperoxidase activity, and inflammatory cytokines (TNF-a, IL-6) [78]. The Fas cell signaling also plays a central role in the physiological regulation of apoptosis. For instance, Bonaccorsi-Riani et al. showed that Fas knockdown by siRNA decreased proinflammatory cytokines (IL-2, CXCX10, TNF-α, and IFN-γ) in a rat liver transplant model with static cold storage [79].

Other mechanisms of ASO activity in the liver include the PI3K/Akt pathway that terminates the BAD-mediated Bcl-2/Bcl-xL antagonism on the mitochondrial membrane and restores their anti-apoptotic function. Chen et al. showed that β-Arrestin-2 (ARRB2) protects hepatocytes against IRI by activating the PI3K/Akt pathway [80]. The administration of a PI3K/Akt inhibitor resulted in severe IRI, whereas ARRB2 knockdown by siRNA significantly reduced PI3K/Akt phosphorylation, inhibited proliferating cell nuclear antigen (PCNA) expression, and increased cleaved caspase-3 expression in H_2_O_2_-treated human hepatocyte cultures. Fujii et al. reported that the Tissue inhibitor of metalloproteinase 3 (TIMP3) protects hepatocytes from undergoing apoptosis by sheltering the E-cadherin/β-catenin complex after IRI [81]. Indeed, TIMP3^−/−^ mice had exacerbated IR-triggered liver damage compared to TIMP3^+/+^ mice, and the knockdown of β-catenin with siRNA in TIMP3^+/+^ hepatocytes increased both caspase 3 and caspase 6 activation. Wang et al. also spotlighted this pathway and applied the HO-1-shRNA (short hairpin RNA) to demonstrate that knockdown promoted hepatocyte pyroptosis in a rat liver transplant, which implies HO-1 is important in hepatic IRI and that it can coordinate with mitochondrial function to enhance the ability of liver cells to resist OxS [82]. These pathways are summarized in Figure 4.

PMOs represent another class of synthetic ASOs that have been used in the study of liver IRI. They consist of a backbone composed of morpholine rings linked by phosphorodiamidate intersubunit linkages, with bases attached to each morpholine ring [83]. They are distinct from other types of ASOs in that while specifically binding to complementary RNA sequences; they prevent translation or splicing of the targeted mRNA by a mechanism involving steric hindrance [84]. Our group recently showed how the alternative splicing of CEACAM1 protects against ischemic liver damage [85]. Owing to the extensive alternative splicing that CEACAM1 undergoes around its variable exon 7, two cytoplasmic tail isoforms are produced, CEACAM1-S (short) and CEACAM1-L (long) [86]. The three RNA splicing factors that have been characterized to control splice-site recognition of exon 7 are Polypyrimidine Tract Binding Protein 1 (PTBP1), Heterogeneous nuclear ribonucleoprotein L (hnRNP L), and Heterogeneous nuclear ribonucleoprotein A1 (hnRNP A1) [85,87,88,89]. PMO’s superior stability, specificity, and minimal off-target effects make them valuable tools for investigating gene function and developing potential therapeutic interventions in various genetic disorders, such as Duchenne muscular dystrophy [90].

In our recent studies, primary mouse hepatocytes cultured with PMOs induced the formation of CEACAM1-S, which consequently protected cells from hypoxia–reoxygenation stress [85]. PTBP1 splicing activity was shown to be coordinated by Hypoxia Inducible Factor 1 Alpha (HIF-1α), a transcription factor stabilized during the cellular response to low oxygen levels [91]. Our retrospective analysis study of human donor liver grafts (n = 46) showed that the cytoplasmic variant *CEACAM1-S* correlated positively with pretransplant levels of *HIF1A* expression levels. This also correlated with better transplant outcomes, including reduced *TIMP1*, total bilirubin, proinflammatory *MCP1* and *CXCL10* cytokines, immune activation marker *IL17A*, and incidence of delayed complications from biliary anastomosis [85].

### 5.2. ASOs Targeting Liver Kupffer Cells

KCs are the primary producers of ROS and other proinflammatory mediators during the initial stage of liver IRI. Recently, Wu et al. discovered that the activation of the stimulator of interferon genes (STING) primarily depends on Kupffer cells (KCs) and that STING facilitates the processing of caspase 1-GSDMD in macrophages, thereby exacerbating liver IRI and cell death through pyroptosis [92]. During this process, STING raises intracellular calcium levels to promote caspase 1-GSDMD processing, which may be associated with ER stress. Applying STING-siRNA successfully confirmed the reduced severity of calcium-dependent macrophage caspase 1-GSDMD-mediated liver IRI in mice. Kong et al. also focused on the cyclic GMP-AMP synthase-stimulator of interferon genes (cGAS-STING) pathway, mainly mediated by KCs. Recognition of aberrant DNA by cGAS leads to activation of cGAS-STING signaling, which triggers the innate immune response. The knockdown of cGAS or STING through siRNA was found to attenuate hepatic IRI-induced inflammation and ameliorate liver function. The authors then used Sirt3-siRNA to show increased post-translational phosphorylation and cGAS/STING activation in hepatocytes, which indicated that inhibiting the cGAS-STING axis is important for reducing hepatic IRI-induced inflammation [93]. In another study, Hu et al. reported that HSP110 in rat liver transplants promotes IRI by activating the Toll-like receptor/Nuclear factor-κB (TLR4/NF-κB) pathway, and this is accompanied by M1 polarization of KCs [94]. M1 macrophages are characterized by their ability to produce pro-inflammatory cytokines, ROS, and NO through the action of inducible nitric oxide synthase (iNOS). HSP110 is part of the family of Heat shock proteins (HSP) that assist in protein folding, assembly, transport, and degradation, ensuring proper protein quality control within cells. The authors showed that HSP110 knockdown reduced the levels of CD86, which plays a role in polarizing M1 macrophages, and this was accompanied by heightened levels of multiple inflammatory cytokines regulated by the NF-κB pathway, such as TNF-α and IL-1β.

NF-κB activation in KCs is also important in the pathogenesis of hepatic IRI by developing inflammation and OxS. Liu and colleagues established that peroxisome proliferator-activated receptor γ (PPARγ), an inflammatory response inhibitor, had a protective effect against liver IRI in mice and hypoxia/re-oxygenated human L02 cells by suppressing the NF-κB signaling. The application of PPARγ-siRNA resulted in greater NF-κB p65 nuclear translocation and increased p-IκBα expression after insult [95]. When gene silencing negatively affects outcomes, combining siRNA and agonists is a tool for experimentation. For example, Zhuang and colleagues employed a G-protein-coupled bile acid receptor (TGR5)-siRNA and a TGR5 agonist (INT-777) to investigate the role of TGR5, also known as GPBAR1 (G protein-coupled bile acid receptor 1), on hepatic IRI. Their findings indicated that TGR5 significantly mitigated liver IRI by activating the Kelch-like ECH-associated protein 1 (Keap1)/Nrf2 pathway [96]. This cellular defense mechanism plays a critical role in maintaining cellular redox balance and protecting against oxidative damage by Nrf2 interactions with Keap1. In response to OxS, ROS modifies cysteine residues on Keap1 and disrupts the interaction with Nrf2 so that it translocates into the nucleus to activate antioxidant response genes [97]. The authors conclude that TGR5 activation in KCs is necessary to activate the Keap1/Nrf2 pathway and, by increasing nuclear Nrf2, this has downstream regulatory activity on HO-1 expression, which is important to enhance cytoprotective anti-IRI mechanism.

### 5.3. ASOs Targeting Other Liver Signaling Pathways

Small ASOs have also been used to study mitochondrial dysfunction and OxS-mediated inflammasome activation in the pathogenesis of NAFLD. The extensive literature on RNA interference shows that small interfering RNAs (siRNAs) are among the most utilized ASO types, owing to their effectiveness in gene silencing through sequence-specific mRNA degradation [98]. In a recent study, Nonsteroidal Anti-inflammatory Drug-Activated Gene-1 (NAG-1) and Growth Differentiation Factor 15 (GDF15) showed significantly lower levels of expression in patients with steatosis as compared to normal control tissue. NAG-1 and GDF15 belong to the transforming growth factor beta (TGF-β) superfamily and play a role in various physiological processes, including inflammation, apoptosis, metabolism, and stress response [99,100]. As a proof-of-concept investigation, siRNAs targeting NAG-1/GDF15 notably exacerbated high-fat diet (HFD)-induced obesity and hepatic steatosis while disrupting lipid homeostasis and nucleic acid exocytosis. Other observations included lower levels of fatty acid β-oxidation and lipolysis but higher fatty acid synthesis and uptake, heightened secretion of IL-18 and IL-1β, and AIM2 inflammasome activation. Importantly, the augmentation of NAG-1/GDF15 expression in transgenic mice substantially ameliorated these characteristics, suggesting a potential role of ROS production and dsDNA release in dampening AIM2 activation by NAG-1/GDF15 under conditions of fatty acid overload [101]. Huang et al. engineered a vitamin A-modified crosslinking nanopolyplex (T-C-siRNA) which effectively targets activated HSCs, the principal cells involved in liver fibrosis, through the controlled release of Platelet-Derived Growth Factor Receptor (PDGFR)-β siRNA upon exposure to ROS and cis diol compounds [102]. This nanopolyplex employs a retinol-binding protein hijacking mechanism to deliver siRNA into HSCs, reducing intracellular ROS levels upon cytoplasmic release. Latorre et al. recently reported that lipopolysaccharide-binding (LBP) protein siRNA (LNP-UNA-si69108) could be utilized to decrease liver lipid aggregation, lipogenesis, and lipid peroxidation-associated OxS markers in mice fed a high-fat and high-sucrose (HFHS) diet to mimic liver steatosis [103]. OxS markers Gsta3, Gpx4, Sod2, and MDA that were enhanced under the HFHS condition were later diminished by LNP-UNA-si69108 injection. Lbp gene silencing by Lbp-specific shRNA mitigated palmitate-induced lipogenesis, OxS, and ER stress and decreased metabolic activity, indicating a protective effect against palmitate-induced liver cell dysfunction. Ariffianto et al. explored the role of the ROS/JNK signaling pathway in the HCV life cycle by investigating its influence on JunB phosphorylation and activation [104]. JunB silencing by siRNA demonstrated that JunB hinders HCV replication in infected cells, indicating its potential as a therapeutic target.

### 5.4. ASOs That Activate Gene Expression in Liver Pathology

The class of ASOs representing a burgeoning frontier in molecular biology studies is the saRNAs [105]. Their superior ability for the precise modulation of gene expression through targeted activation of specific genes at the site of promoter structures makes them ideally suited for therapeutic applications for neurodevelopmental disorders [106], cancer treatment [107], and diabetes [108]. Acting as short double-stranded RNA fragments, they are processed by the same RISC complex as siRNAs (Figure 2) [109]. Most recently, saRNAs were employed to investigate the function of CCAAT enhancer-binding proteins (CEBPs) as pivotal transcriptional regulators crucial for preserving liver functionality in Hepatocellular carcinoma (HCC) [110]. Another group utilized SIRT1 saRNAs, a member of the NAD+-dependent deacetylase family, to reduce inflammatory-like responses and re-establish normal lipid metabolism. SIRT1 regulates various cellular processes, including gene expression, DNA repair, metabolism, and aging [111]. Elevated expression resulted in decreased mRNA levels of pivotal inflammatory cytokines, including Tumor Necrosis Factor-α (TNF-α), Interleukin 1β (IL-1β), and keratinocyte chemoattractant, alongside notable reductions in the phosphorylation of NF-κB and c-Jun N-terminal kinase, pivotal molecules in inflammatory signaling pathways. In the group treated with SIRT1 saRNA, animals exhibited substantially reduced weight gain, decreased white adipose tissue mass, lowered triglyceride levels, reduced fasting glucose levels, and diminished intracellular lipid accumulation.

Thus far, the ASOs discussed have fallen into a single “activator” or “silencer” category. However, LNAs, specifically gapmers and mixmers, present relatively new and upcoming ASOs in the field for their ability to induce loss-of-function or gain-of-function expression changes [112,113]. In addition to their versatility, LNAs work with much greater specificity and stability in targeting a gene and do not require a delivery agent to enter the cell. These unique properties of LNAs arise from their “locked” structure (Figure 2). LNAs have a bridge connecting the 2′-oxygen to the 4′-carbon in the ribose sugar structure, creating a locked and stiff ribose group. This characteristic gives LNAs the ideal structure as a nucleotide analog, able to bind and hybridize DNA and RNA with a far greater complementarity to exert its effects. The structure of LNAs also gives rise to increased thermal stability as well as better solvation and ease of delivery due to the more lipid-soluble nature of the 2′-oxygen to the 4′-carbon methyl linkage, allowing it to forgo any potentially toxic delivery mechanisms [114]. Within the category of LNAs, mixmers include DNA and LNA nucleotides randomly interspersed within the oligonucleotide, while gapmers have two sequences of LNA nucleotides surrounding a DNA sequence within the oligonucleotide [115]. Gapmers exert their effects primarily through RNAse H activity, causing nucleolytic cleavage to stop translation, while mixmers work by sterically blocking ribosome binding, preventing the addition of the 5′ cap [116].

Recently, LNAs were used to target Vasohihibin-2 (VASH2), a highly conserved gene overexpressed in cancers [117]. When systemically administered, naked 2′,4′-bridged nucleic acids (BNA)-based VASH2-ASO localized to the liver and, importantly, its gene-silencing function showed potent antitumor activity on human HCC cells. Another group utilized gapmers to improve outcomes for patients suffering from Ullrich congenital muscular dystrophy (UCMD) [118]. Gain-of-function mutations in the COL6A gene are implicated in UCMD, and these researchers were able to design a gapmer to target the dominant mutant allele of the COL6A gene, thereby silencing this mutation and increasing levels of the functional COL6 gene, ameliorating UCMD. Despite virtually no literature on using saRNAs and LNAs in the majority of liver disorders (e.g., including NAFLD, NASH, cancer, and drug-induced liver toxicity), the collective studies in other fields underscore the remarkable progress made in understanding the potential of ASOs in various fields and present a significant opportunity for OxS mitigation using RNA-based therapeutics.

## 6. ASOs in Recent Clinical Studies

A significant caveat in support of building a therapeutic ASO armamentarium for treating liver pathologies lies in the fact that many clinical trials using ASOs are currently underway in areas as diverse as splicing modulation [119] and the treatment of muscular [120] and metabolic disorders [121]. A literature search using the keywords “antisense oligonucleotides and clinical trials,” filtered for years 2023–2024, yielded 15 studies that used ASOs to treat the nervous system (20%, 3/15), the muscular and cancer systems (both 13.3%, 2/15), or the immune system (6.7%, 1/15). By contrast, 46.7% (7/15) of the clinical trials targeted the liver, and among the total 15, 20% of liver studies (3/15) were designated Randomized Control Trials (RCT), as compared to only one RCT study (6.7%) among all other organ systems. Among the RCT studies targeting the liver, GalNAc (N-Acetyl galactosamine) ASO conjugation was used in patients with chronic hepatitis B [11]. This modification involves attaching GalNAc molecules to the oligonucleotide, typically via a linker, which then facilitates specific binding to cell surface receptors known as asialoglycoprotein receptors (ASGPRs) primarily located on hepatocytes in the liver [122]. In one mouse study, the targeted delivery of ASOs to hepatocytes using Gal-Nac structures improved the potency of targeting human apolipoprotein C-III and human transthyretin (TTR) 10-fold in mice [123]. The authors showed that plasma pharmacokinetics, their primary endpoint analysis, was not significantly altered in their Asia-Pacific population after using GSK3389404, a GalNac-conjugated ASO [11]. Two additional clinical trials were undertaken utilizing GalNAc3-conjugated 2′-O-methoxyethyl (2′MOE) ASOs in hepatic applications, demonstrating the well-tolerated nature of ASOs with no discernible class effect observed across all doses administered when compared to placebo [16,17]. Yeang et al. investigated whether ASO pelacarsen showed activity against low-density lipoprotein cholesterol (LDL-C) and lipoprotein(a) cholesterol content [124]. Pelacarsen, formerly known as Inclisiran, is a novel ASO RNA-based therapeutic agent used to treat hypercholesterolemia, specifically to lower low-density LDL-C levels. It works by blocking the liver production of proprotein convertase subtilisin/kexin type 9 (PCSK9) [125]. Finally, there is much excitement regarding Bepirovirsen (GSK3228836), a 2′-O-methoxyethyl modified antisense oligonucleotide, in treating chronic hepatitis B [126,127]. These studies are summarized in Table 1. Collectively, targeting liver pathologies shows a well-established foundation laying the groundwork for the therapeutic application of ASOs in OxS-related hepatocellular injuries in future clinical investigations.

## 7. Challenges, Future Directions, and Conclusions

Moving forward, our aging population and continued exposure to our Western diet ensures that liver diseases will become more widespread and remain a clinical challenge. The use of ASOs is in its infancy and will attract much attention in future studies. For the moment, ASO therapies may be better suited for monogenic diseases, where a mutation in a single gene is responsible for the disease phenotype. Correcting or modulating the expression of an mRNA transcript may be a more parsimonious approach, leading to selective inhibition of the abnormal gene product. By contrast, cardiovascular diseases, diabetes, and many types of cancer are examples of multifactorial diseases where the challenge of controlling for multiple genetic variants, environmental factors, and complex biological pathways may pose undue technical challenges. Combinatorial approaches involving ASOs and other therapeutic modalities, such as small pharmacological molecules or immunomodulators, may be an ideal approach that synergistically enhances treatment outcomes while overcoming resistance mechanisms. A series of recent studies demonstrated the advantages and risks of using ASO combinatorial therapy. When SMN2 (Survival Motor Neuron 2) and Neurocalcin delta (NCALD) ASOs were tested in a severe spinal muscular atrophy (SMA) mouse model, significant amelioration of histological and electrophysiological SMA hallmarks were observed at postnatal day 21 [128]. By contrast, in other studies using combinatorial ASO-mediated therapy in the SMA model using low doses of SMN and the protective modifier Chp1 calcineurin like EF-hand protein 1 (CHP1), no notable enhancement in SMA hallmarks at 2 months of age, raising the precaution against using ASOs that are short-lived [129]. Thus, the concentration of ASO’s, targeting potential and persistence must be optimized to ensure low drug toxicity vs. high biological effect.

The fidelity of maintaining the endocytic pathway, the primary means that small molecules enter the cell, is another notable consideration for using ASO therapy. Regulating the proteins that control the endocytic pathway may unexpectedly cause disruptions in cellular processes beyond the intended targets, leading to off-target effects. These effects can manifest as alterations in membrane dynamics, impaired receptor recycling, or unintended changes in intracellular signaling cascades. Consequently, careful consideration of the interconnected nature of cellular pathways is essential when modulating endocytic protein activity to minimize unintended consequences and ensure therapeutic efficacy. Some proteins that are known to regulate the endocytic pathway include clathrin, which forms a scaffold around the vesicle during its formation, and adaptor proteins such as AP2, which mediate the recruitment of cargo molecules into the vesicle [130,131]. Additionally, small GTPases like Rab proteins regulate vesicle trafficking and fusion with target membranes, ensuring molecules’ efficient and regulated transport within the cell [132].

Alternatively, delivering ASOs to specific cell types via targeted endocytosis remains a challenge that needs more attention. In the liver, the use of Gal-Nac-coupled ASO represents a significant advance, but for other organ systems, the future challenge will be to identify cell surface target receptors that allow ASO vacuolization, especially considering the inefficiencies of the endosomal pathway [133]. Studies have shown that conjugating ASOs to glucagon-like peptide 1 (GLP1) enhances targeted knockdown in pancreatic β-cells [134] and that targeting coat protein complex II (COPII) vesicles, typically involved in ER-Golgi transport, can re-locate ASOs to late endosomes (LEs) vesicles upon incubation [135]. In contrast, the activation of autophagy was demonstrated to be crucial for the improved localization of ASOs within autophagosomes without affecting intracellular concentrations or trafficking to alternative compartments [136]. Thus, future studies will need to focus on the molecular factors needed to optimize endosomal escape to enhance our understanding of why different cell types control ASO uptake differently before the pharmacological effects of ASOs are fully realized.

Similarly, precision medicine approaches incorporating patient-specific genetic profiles and disease characteristics could optimize ASO therapies by selecting appropriate targets and treatment regimens tailored to individual patients. For example, in Familial Amyotrophic Lateral Sclerosis (FALS), which is responsible for up to 15% of ALS cases, the recent use of subpial injections of Adeno-Associated Virus serotype 9 (AAV9)+anti Superoxide Dismutase 1 (SOD1) ASOs prevented the disease in animal models [116]. Although viral vectors in humans are limited by their immunogenicity, tailoring medical treatments and interventions based on a patient’s genetic makeup, lifestyle, and environment remains a rapidly evolving and promising area of future research studies.

While the use of ASOs in OxS-related liver injuries and disease is limited, existing studies in other fields have provided valuable insights. They should pave the way for the potential implementation of ASOs into current treatment protocols for organ IRI and other metabolic diseases. Continued advancements in delivery technologies, such as lipid nanoparticles, conjugates, or viral vectors, promise to improve the pharmacokinetics, tissue specificity, and safety profiles of ASOs in liver therapeutics. Modifying donor livers with adjunctive ASOs during ex vivo machine perfusion represents an exciting opportunity to rejuvenate otherwise marginal or discarded organs. Such customized, individualized gene modification strategies in transplant organs should maximize both graft and recipient survival, minimize the acute shortage of livers, and increase the donor pool available for life-saving transplant procedures. By leveraging the knowledge gleaned from diverse fields, we can accelerate the translation of ASO-based interventions into effective treatments for liver disorders characterized by OxS.

## Figures and Tables

**Figure 1 antioxidants-13-00678-f001:**
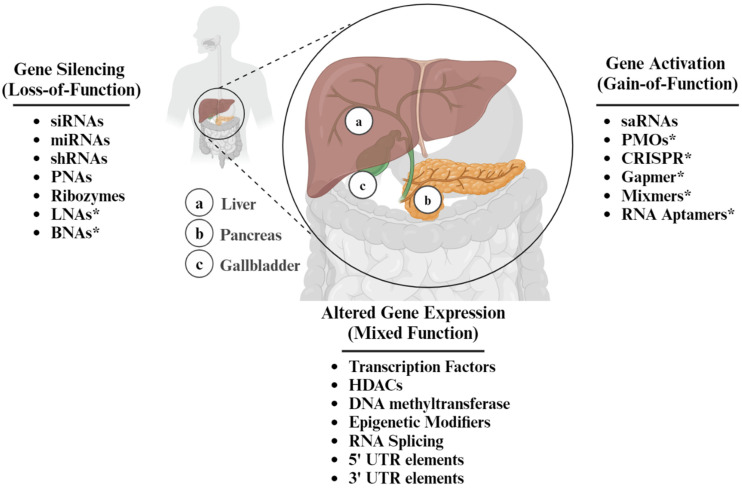
Overview of gene expression modulating therapeutic strategies. Asterisk (*) denotes strategies that cause gene silencing or activation, depending on the cellular context. UTR is Untranslated region.

**Figure 2 antioxidants-13-00678-f002:**
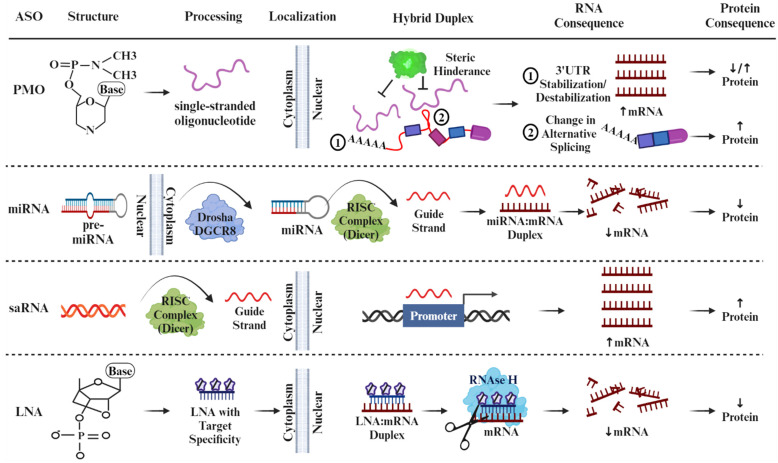
Diagram depicts cellular uptake mechanisms for ASOs that result in productive and nonproductive end-products. Up and down arrows indicate the direction of expression, either higher or lower.

**Figure 3 antioxidants-13-00678-f003:**
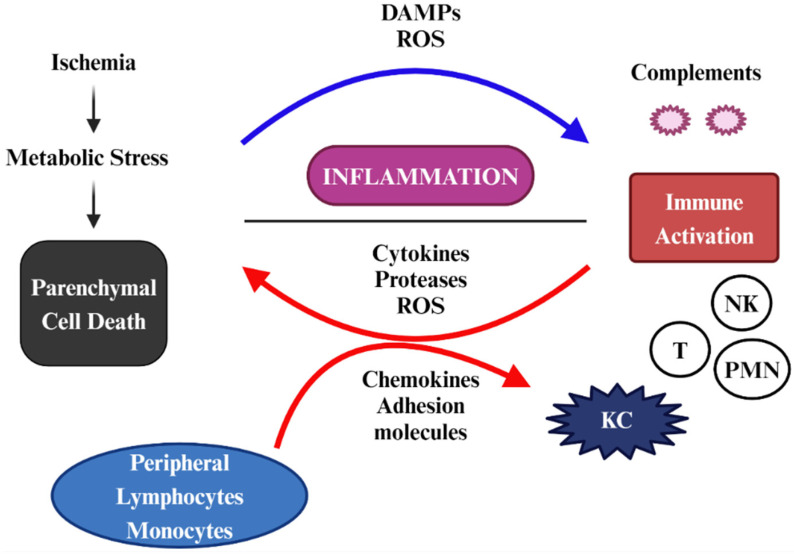
Liver ischemia-reperfusion injury (IRI) occurs in distinct stages. IRI is a localized process that causes hepatic metabolic disturbances. It arises from glycogen consumption, oxygen deprivation, and adenosine triphosphate (ATP) depletion. The release of DAMPs from necrotic cells, tissue injury-induced complement activation, and mitochondrial ROS production triggered by reoxygenation contribute to liver immune activation following reperfusion. This involves various liver nonparenchymal cell types, including KCs, dendritic cells, T cells, natural killer (NK) cells, and neutrophils. The ischemia-reperfusion (IR)-initiated local proinflammatory cascade perpetuates itself by recruiting peripheral immune cells from the circulation, ultimately leading to liver failure. Adapted from [35] and created by Biorender.com.

**Figure 4 antioxidants-13-00678-f004:**
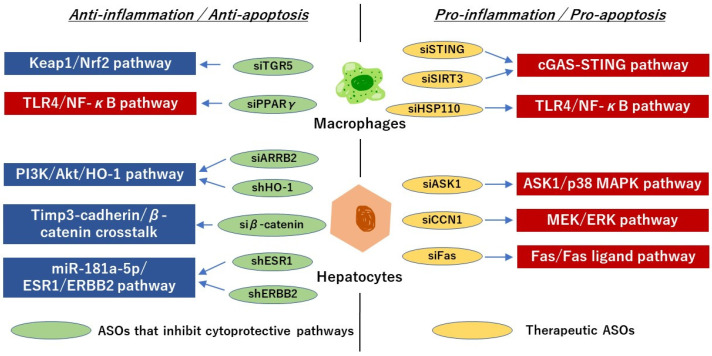
Newly reported RNA-based small molecule agents and associated signaling pathways of IRI. Abbreviations: Apoptosis signal-regulating kinase 1 (ASK1), β-Arrestin-2 (ARRB2), Cellular communication network factor 1 (CCN1), Cyclic GMP-AMP synthase (cGAS), v-erb-b2 avian erythroblastic leukemia viral oncogene homolog 2 (ERBB2), Extracellular signal-regulated kinase (ERK), Estrogen receptor 1 (ESR1), Heme oxygenase-1 (HO-1), Heat shock protein 110 (HSP110), Kelch-like ECH-associated protein 1 (Keap), Mitogen-activated protein kinase MAPK MAPK/ERK kinase (MEK), Nuclear factor erythroid 2–related factor 2 (NRF2), Phosphoinositide 3-kinases (PI3K), Peroxisome proliferator-activated receptor γ (PPARγ), short hairpin RNA (shRNA), Stimulator of interferon genes (STING), Tissue inhibitor of metalloproteinase 3 (Timp3), Toll-like receptor 4 (TLR4).

**Table 1 antioxidants-13-00678-t001:** Representative clinical studies targeting ASOs to organ systems, collected from years 2023–2024.

Disease	Therapy	Study Design Phase, N, Year	Novelty	Outcome	Source
Cancer					
NSCLC *	Danvatirsen αSTAT3 ASO	CT *, P2 *, 83, 2023	Combination regimen using PD-(L)-1 inhibition with other agents, ASOs	Preliminary efficacy signals showed no major pathologic response (MPR) as the primary endpoint	[9]
NSCLC		CT, P2, 268, 2024		Combination therapy with Danvatirsen did not improve immunosuppressive TME * targeting	[10]
Liver					
CHB *	GSK3389404	RCT, 64, P2, 2023	GalNac targeting to parenchymal hepatocytes	Safety study showing that plasma pharmacokinetics from any Asia-Pacific population may be used to guide ASO dose selection	[11]
Lp(a) *	Pelacarsen	RCT, 29, P1, 2023	Safety and efficacy in Japanese populations studied	No clinically relevant abnormalities were detected	[12]
HBV *	Bepirovirsen GSK3228836	RCT, 440 P2b, 2023	Efficacy and safety study	Assessment of hepatitis B surface antigen and DNA seroclearance levels may be discontinued in the presence and absence of background nucleos(t)ide analogue therapy	[13]
Dyslipidemia	Vupanorsen (PF-07285557)	CT, P1/2, 451, 2023	2nd generation ligand-conjugated 2′O-methoxyethyl modified angiopoietin-like 3	ANGPTL3 target reduction of 75% achieved with a 320-mg dose of Vupanorsen per month	[14]
Dyslipidemia		CT, P1, 6, 2023		Vupanorsen reduces triglycerides, lipids, and apolipoproteins.	[15]
No Disease	GalNAc_3_ *-conjugated 2′MOE * ASOs	CT, P2, 195, 2023	Safety and Efficacy Study	No class effect was identified, ASOs were well-tolerated in all doses tested compared to controls	[16]
No Disease		CT, P1, 195, 2024	GalNac targeting to parenchymal hepatocytes	Safety and tolerability observed in GalNac-conjugated than unmodified ASOs	[17]
Immune					
HAE *	Donidalorsen ligand-conjugated antisense LICA	CT, P2, 20, 2024	Targets the prekallikrein (PKK) pathway	No adverse effects with Donidalorsen. Showing durable efficacy with 96% less HAE attacks	[18]
Muscular					
MDT1 *	Baliforsen (ISIS 598769)	CT, P1/2, 20, 2023	ASO targeting DM1 protein kinase (DMPK) mRNA	Baliforsen was well-tolerated but below levels predicted to achieve substantial target reduction—more studies will be needed	[19]
DMD *	ATL1102	CT, P2, 9, 2024	2′MOE gapmer antisense oligonucleotide to the CD49d alpha subunit of VLA-4	Multiple muscle disease progression parameters assessed show stabilization and safety in non-ambulant boys with DMD	[20]
Nervous					
AD *	MAPTRx	CT, P1b, 46, 2023	Inhibition of MAPT expression with a Tau-targeting ASO	Reduction in CSF total tau concentration with 50% mean reduction 24 weeks following last dose	[21]
AD *	BIIB080	RCT, P1b, 46, 2023	a MAPT-targeting antisense oligonucleotide	BIIB080 reduced tau biomarkers in study subjects with mild AD	[22]
ATTRv *	Inotersen	CT, P3, 172, 2023	Inhibits the production of transthyretin (TTR) protein	Inotersen led to lower muscle weakness measures following 65 weeks treatment as compared to placebo controls	[23]

* Abbreviations: Alzheimer’s disease (AD), Cerebrospinal fluid (CSF), Chronic Hepatitis B Infection (CHB), Clinical Trial (CT), Duchenne muscular dystrophy (DMD), Hepatitis B virus (HBV), Hereditary angioedema (HAE), Hereditary transthyretin amyloidosis (ATTRv), Lipoprotein(a) (Lp(a)), Microtubule-Associated Protein Tau (MAPT), Myotonic dystrophy type 1 (MDT1), Non-small Cell Lung Cancer (NSCLC), Phase (P), Programmed Death-Ligand 1 (PD-(L)-1), Random Control Trial (RCT), Triantennary N-acetylgalactosamine (GalNAc3), 2′-O-methoxyethyl (2′MOE), Tumor microenvironment (TME).

## Data Availability

Not applicable.

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
