# Peer review of "The Coming Age of Antisense Oligos for the Treatment of Hepatic Ischemia/Reperfusion (IRI) and Other Liver Disorders: Role of Oxidative Stress and Potential Antioxidant Effect"

_antioxidants, 2024, doi:10.3390/antiox13060678_

Round 1

Reviewer 1 Report

The review by Yao et al. focuses on novel strategies to modulate transcription and translation in liver disorders using antisense oligo (ASO) -based therapies. This work is very well-written, with respect to English usage, generally well-organized with some exceptions, and contains nice and helpful illustrations and many recent references. Nevertheless, this review is only loosely related to oxidative stress (OS) and antioxidants, since most of ASO discussed in the review do not target OS directly, but different signaling pathways that may indirectly influence the cellular redox balance. In addition, most of liver disorders, and most of other diseases are somehow associated with OS. Therefore, the term of “OS Liver-Related disorders” proposed by the authors is exaggerated and artificial. Besides that, the content of some paragraphs needs to be improved, as they are sometimes confusing or missing some important information.  These and other problems are pointed out below.

Major issues:

1.     The authors discuss IRI and the role of oxidative stress in many other liver diseases including NAFLD, NASH, cancer and drug-induced liver toxicity, suggesting that OS is a common denominator of the majority of liver disorders.  For this reason, there is no need to distinguish a group of “OS-stress related liver injuries/diseases” and this term should be eliminated from the title and from the review in general.

2.     For the same reason, the reference to OS in the title should be changed or removed. For example: “The Coming Age of Antisense Oligos for the Treatment of Hepatic Ischemia/reperfusion (IRI) and Other Liver Disorders: role of oxidative stress and potential antioxidant effects”

3.     The paragraph 3 is confusing and denotes rather fragmentary knowledge of the role of oxidative stress in liver diseases. It should be reorganized, and other important recent papers should be cited. First of all: it has NOT been recently investigated, but known as key pathogenic factor in the progression of many liver diseases. The authors may refer to a recent review by Chen, Z., et al., Role of oxidative stress in the pathogenesis of nonalcoholic fatty liver disease. Free Radic Biol Med, 2020. or by Seen S. et al. Chronic liver disease and oxidative stress”. Expert Rev Gastroenterol Hepatol. 2021.

4.     Another problem is that review misses a critical analysis of safety and efficacy of RNA-based therapies, of liver diseases. The authors should add another full paragraph and discuss the pro- and cons- of RNA-based therapies, including challenges associated with off-target side effects and insufficient biological activity.

Minor issues:

1.      Abstract. The expression in line 17 “small molecule inhibitors/activators that control the expression of enzymes, transcription factors, … etc.” is confusing for the readers as it suggests that a range of different not RNA-based molecules compounds has been reviewed, beside ASO.

2.      Table 1 lists clinical trials reporting on the efficacy and safety of ASO-based therapies, therefore the name of the first column should not be “Model”, but “Disease”.

3.      In the same column of the Table 1 “hepatocytes” should be substituted with “no disease” or “healthy”.

4.      Table 1, column: Novelty, line 2:  PD-(L)-1, this abbreviation should be explained below the table.

5.      Introduction. It misses the definition and description of different ASO types. Thus the first part of the chapter 4 “The duality of ASOs as a therapeutic intervention“ should be moved to  the Introduction.

6.      Introduction: Linde 44: the statement “the liver is particularly accessible for the development of novel …” is unclear and it should be well explained.

7.      The title of the chapter 5 should be rephrased.  “Role of ASOs in signaling pathways involved in liver IRI” to “Modulation of signaling pathways involved in liver IRI by ASO”.

Author Response

Dear Reviewer 1,

We thank you for your careful reading of our MS. We have addressed all your inquiries and responded to each point in Yellow or blue indicating where new text has been added to the MS.

================

Reviewer 1

The review by Yao et al. focuses on novel strategies to modulate transcription and translation in liver disorders using antisense oligo (ASO) -based therapies. This work is very well-written, with respect to English usage, generally well-organized with some exceptions, and contains nice and helpful illustrations and many recent references. Nevertheless, this review is only loosely related to oxidative stress (OxS) and antioxidants, since most of ASO discussed in the review do not target OxS directly, but different signaling pathways that may indirectly influence the cellular redox balance. In addition, most of liver disorders, and most of other diseases are somehow associated with OxS. Therefore, the term of “OxS Liver-Related disorders” proposed by the authors is exaggerated and artificial. Besides that, the content of some paragraphs needs to be improved, as they are sometimes confusing or missing some important information.  These and other problems are pointed out below.

Major issues:

  1. The authors discuss IRI and the role of oxidative stress in many other liver diseases including NAFLD, NASH, cancer and drug-induced liver toxicity, suggesting that OxS is a common denominator of the majority of liver disorders.  For this reason, there is no need to distinguish a group of “OxS-stress related liver injuries/diseases” and this term should be eliminated from the title and from the review in general.

Three mentions of “OxS-stress related liver injuries/diseases” were deleted (Line 27, Line 341, Line 346 of the original MS). Next text on line 422 was added (e.g., including NAFLD, NASH, cancer and drug-induced liver toxicity)

  1. For the same reason, the reference to OxS in the title should be changed or removed. For example: “The Coming Age of Antisense Oligos for the Treatment of Hepatic Ischemia/reperfusion (IRI) and Other Liver Disorders: role of oxidative stress and potential antioxidant effects”

Response:

The title was changed to “The Coming Age of Antisense Oligos for the Treatment of Hepatic Ischemia/Reperfusion (IRI) And Other Liver Disorders: Role of Oxidative Stress and Potential Antioxidant Effect”, per Reviewer recommendation

  1. Paragraph 3 is confusing and denotes rather fragmentary knowledge of the role of oxidative stress in liver diseases. It should be reorganized, and other important recent papers should be cited. First of all: it has NOT been recently investigated but known as key pathogenic factor in the progression of many liver diseases. The authors may refer to a recent review by Chen, Z., et al., Role of oxidative stress in the pathogenesis of nonalcoholic fatty liver disease. Free Radic Biol Med, 2020. or by Seen S. et al. Chronic liver disease and oxidative stress”. Expert Rev Gastroenterol Hepatol. 2021.

Response:

Paragraph 3 which is labeled “3. OxS in other liver diseases” has been reorganized with more general information on NAFLD. We also added that OxS is a key pathogenic factor in liver diseases, as recommended by the Reviewer. We have also added more references and included the Chen et al. and Seen et al reports. For uniformity reasons, we have changed all instances of oxidative stress (OS) to (OxS), following the convention established in the Chen et al, 2021 report. Lines 114 onward.

New Text

OxS is key pathogenic factor in the progression of many liver diseases, such as non-alcoholic fatty liver disease (NAFLD), an increasingly common worldwide condition that is asymptomatic in its early stages (15, 16). NAFLD is actually a spectrum of liver conditions ranging from simple steatosis (fatty liver) to non-alcoholic steatohepatitis (NASH), which involves inflammation and liver cell damage, and can progress to fibrosis, cirrhosis, and even liver cancer in severe cases (17). Studies show that patients with NAFLD exhibit higher levels of OxS and lipid peroxidation products in their serum/plasma blood fluid (18). ROS typically found in NAFLD patients derive from mitochondrial generated superoxide anions (O2•−), which are byproducts of oxidative phosphorylation, and peroxisomes, which function by breaking down long-chain fatty acids in a process called beta-oxidation. As chronic liver injury progresses, protective antioxidant defense mechanisms fail to overcome the induction of OxS-sensitive transcription factors, such as NF-kB, Egr-1, and AP-1 that ultimately lead to hepatocyte cell death (19, 20).

Emerging evidence suggests a role for cellular networks that crosstalk with our immune cells and gut microbiota in facilitating chronic liver disease progression. For example, a recent study on NAFLD and cirrhosis described a subpopulation of human resident liver myeloid cells (LM) that were protective against obesity-associated OxS development (21). LMs were shown to upregulate Peroxiredoxin 2 (PRDX2), a biological catalyst that reduces hydrogen peroxide, organic hydroperoxides, and peroxynitrite, essential for detoxifying harmful compounds. Though functional validation was demonstrated using human 2D and 3D cultures, it remains to be seen whether LMs alleviate the OxS load once metabolic diseases linked to obesity have been initiated.

  1. Another problem is that review misses a critical analysis of safety and efficacy of RNA-based therapies, of liver diseases. The authors should add another full paragraph and discuss the pro- and cons- of RNA-based therapies, including challenges associated with off-target side effects and insufficient biological activity.

We have added new text to Chapter 4 that addresses Reviewer comments, line 199

The versality of ASOs that make them good candidates for therapeutic application stems from the high degree of sequence specificity that leads to the degradation, modulation, or manipulation by alternative splicing of target RNAs (38). Their combined sensitivity and specificity to both enhance and reduce protein expression sets them apart from my siRNA or microRNAs, that are generally restricted to silencing targeted expression (39). Another advantage of ASOs is their relatively straightforward design and production, so the reduced time between conceptualization and clinical use offers the possibility of rapid development of patient-customized treatments. In one promising example, the development of Milasen, a splice-modulating antisense oligonucleotide drug, was designed and tested 1 year after first contact with a single six-year-old patient with Batten disease, a rare, fatal, inherited disorder of the nervous system (40). ASOs solve the intractable challenge of targeting RNA transcripts of genes that are considered "undruggable" by conventional approaches, offering a clinician an armamentarium that treats a wider range of genetic disorders (41). Other significant advantages include long-lasting effects, minimal immune response, and application across various tissues. This was demonstrated recently in a study of dystrophia myotonica type 1 (DM1), a multi-systemic genetic disorder characterized by progressive muscle wasting and weakness (42). ASOs (IONIS 486178 ASO) delivered to the central nervous system led to a 30–50% reduction of human Dystrophia Myotonica Protein Kinase (DMPK) mRNA 12 weeks after injection in mice.

The cellular uptake of ASOs as a therapy are not without their own challenges and limitations. First, the hydrophobic nature of the phospholipid membrane may hinder the ability of ASOs to target RNA molecules. A strategy using chemical ligation of palmitate, tocopherol and cholesterol to plasma proteins such albumin and lipoproteins has been shown to be effective in targeting extra-hepatic tissues in mice (43). Similarly, endosomal entrapment may inadvertently lead to nuclease digestion susceptibility. To overcome this, studies show that OECs (oligonucleotide enhancing compounds), such as sodium butyrate, are effective at perturbating multivesicular bodies (MVB), a type of endosome involved in the sorting and trafficking of cellular components (44, 45). Sodium butyrate is a histone deacetylase (HDAC) inhibitor that can enhance the efficacy of antisense oligonucleotides (ASOs) and other oligonucleotide-based therapies by improving their uptake and activity within cells (46, 47). Unmethylated CpG motifs may elicit immune responses that lead to adverse effects or decreased efficacy. These considerations may also lead to off-target effects by ASOs that inadvertently interact with unintended RNA targets that lead to undesirable biological consequences and require time-consuming optimization in the clinical setting. Finally, the challenge of choice of administration route (e.g., systemic injection, local injection, or oral administration) as well dose optimization may be needed to reduce the chance of hepato- and renal toxicity, with each route presenting unique challenges and considerations.

Detail comments

Minor issues:

  1. Abstract. The expression in line 17 “small molecule inhibitors/activators that control the expression of enzymes, transcription factors, … etc.” is confusing for the readers as it suggests that a range of different not RNA-based molecules compounds has been reviewed, beside ASO.

We changed the text to say, “we review ASO inhibitors/activator strategies to modulate transcription and translation using that control the expression of enzymes”, as recommended (line 27).

  1. Table 1 lists clinical trials reporting on the efficacy and safety of ASO-based therapies, therefore the name of the first column should not be “Model”, but “Disease”.

We have changed Model to Disease as recommended.

  1. In the same column of the Table 1 “hepatocytes” should be substituted with “no disease” or “healthy”.

We have changed Hepatocytes to No Disease as recommended.

  1. Table 1, column: Novelty, line 2:  PD-(L)-1, this abbreviation should be explained below the table.

We have added Programmed Death-Ligan/d 1 (PD-(L)-1) below Table 1 as recommended

  1. Introduction. It misses the definition and description of different ASO types. Thus the first part of the chapter 4 “The duality of ASOs as a therapeutic intervention“ should be moved to  the Introduction.

We have moved the first part of Chapter 4 to the Introduction as recommended. We also reordered the Figures 1-3 to accommodate this change in the text. In place of this text, we have added a paragraph addressing off-target effects and challenges of using ASOs.

  1. Introduction: Linde 44: the statement “the liver is particularly accessible for the development of novel …” is unclear and it should be well explained.

We have added text to clarify this point. The liver regulates energy and lipid metabolism and, as part of the body’s central metabolic organ, it has potent immunological functions. Line 57

  1. The title of chapter 5 should be rephrased.  “Role of ASOs in signaling pathways involved in liver IRI” to “Modulation of signaling pathways involved in liver IRI by ASO”

We have changed the text for Chapter 5, as recommended.

Reviewer 2 Report

The manuscript reviews strategies to modulate transcription and translation in OS-stress-related liver injuries using small molecule inhibitors/activators that control the expression of enzymes, transcription factors, and intracellular sensors of DNA damage. It well organized and written paper. The topic is up to date.  Before pubIication please address to the following comments:

1. In lines 75-78 you wrote: "Liver IRI generated superoxides (O2•), the hydroxyl radical (•OH), peroxynitrite 76 (ONOO-), and nitrogen dioxide (•NO2), trigger the production of even more reactive radicals" Please give examples of "even more reactive radicals'

 2. In the Chapter 3 , lines 113-115 "Interestingly, recent research suggests that disturbances in gut microbiota,  such as dysbiosis, may play a role in OS, exacerbating the adverse effects of valproate  (VPA) treatment (16)." Reading further description of the study (ref. 16)please add which bacteria strains were responsible for downregulation of CAT, GST, SOD, and HO-1, and upregulation of CYP2E1

3. In the second paragraph of Chapter 3, authors meantioned about potential of using antioxidant compounds to mitigate OS-related liver damage. However, they described only verbenalin. There are numerous compounds that have antioxidant properties such as metformin or vitamin A, C, E or resveratrol that can reduce oxidative stress in the liver. Metformin, a first line drug in type 2 diabtets mellitus therapy,  improves sensitivity of  peripheral tissues to insulin, including liver. Thus, since type 2 diabetes mellitus and NAFLD coexists, metformin exerts beneficial effect in NAFLD patients. Please comment metformin in NAFLD, and add other compounds possessing antioxidant properties applied in NAFLD therapy.

4. What is authors' opinion about the utility of ASOs in the treatment monogenic diseases and multifactorial diseases. Namely, for what types of diseases, ASOs seem to be more effective based ob results of studies described in Chapter 5 and 6. Please comment.

1. Provide the full name of used abbreviations:  "PD-(L)1, MAPT" in Table 1 

2. In line 142  shortcut "8-OHG"  was applied, please provide its full name.

Author Response

Dear Reviewer 2,

We thank you for your careful reading of our MS. We have addressed all your inquiries and responded to each point in Yellow or blue, indicating where new text has been added to the MS.

================

Reviewer 2

The manuscript reviews strategies to modulate transcription and translation in OxS-stress-related liver injuries using small molecule inhibitors/activators that control the expression of enzymes, transcription factors, and intracellular sensors of DNA damage. It well organized and written paper. The topic is up to date.  Before publication please address to the following comments:

  1. In lines 75-78 you wrote: "Liver IRI generated superoxides (O2•), the hydroxyl radical (•OH), peroxynitrite 76 (ONOO-), and nitrogen dioxide (•NO2), trigger the production of even more reactive radicals" Please give examples of "even more reactive radicals'.

Response:

We have added new text and citations to clarify. “During hepatic IRI, enzymes like xanthine oxidase and NADPH oxidase in the mitochondria initiate the production of ROS (12). An inflammatory cascade, termed nitro-oxidative stress, follows with inducible nitric oxide synthase (iNOS) leading to nitric oxide (NO) build-up and in the presence of O2•, highly reactive peroxynitrite (ONOO-) forms (13). In many cases of pathological IRI, peroxynitrite can be further converted to even more reactive radicals, such as nitrogen dioxide (•NO2) (14).”Line 93

  1. In the Chapter 3 , lines 113-115 "Interestingly, recent research suggests that disturbances in gut microbiota,  such as dysbiosis, may play a role in OxS, exacerbating the adverse effects of valproate  (VPA) treatment (16)." Reading further description of the study (ref. 16)please add which bacteria strains were responsible for downregulation of CAT, GST, SOD, and HO-1, and upregulation of CYP2E1

We have added new text, per Reviewer recommendation. “Analyses of bacteria from the VPA group that induced hepatic liver steatosis showed higher levels of Actinobacteriota, Acidobacteriota and Gemmatimonadota, whereas probiotic treatment significantly reversed the changes.” Line 144

  1. In the second paragraph of Chapter 3, authors mentioned about potential of using antioxidant compounds to mitigate OxS-related liver damage. However, they described only verbenalin. There are numerous compounds that have antioxidant properties such as metformin or vitamin A, C, E or resveratrol that can reduce oxidative stress in the liver. Metformin, a first line drug in type 2 diabtets mellitus therapy,  improves sensitivity of  peripheral tissues to insulin, including liver. Thus, since type 2 diabetes mellitus and NAFLD coexists, metformin exerts beneficial effect in NAFLD patients. Please comment metformin in NAFLD, and add other compounds possessing antioxidant properties applied in NAFLD therapy.

New text was added following recommendations.

Other compounds that reduce OxS in the liver include Metformin, and vitamin A, C, E. As an allosteric regulator of mitochondrial glycerophosphate dehydrogenase (mGPD), Metformin alters the balance of NADH and NAD+ within the cell, leading to reduced conversion of lactate and glycerol to glucose, which in turn leads to decreased hepatic gluconeogenesis (32). Studies have shown Metformin improves sensitivity of peripheral tissues to insulin, including the liver, making it a first line drug in type 2 diabetes mellitus therapy (33). A recent study investigated whether adipose mesenchymal stem cell derived exosomes (ADSCs-Exo), functioning as a vehicle to deliver Metformin (Met), would provide a mitochondrial protective role in the treatment of hepatic IRI (34). The study showed that the application of ADSCs-Exo in vivo was effective in inhibiting mitochondrial fission-related protein expression through the AMPK (AMP-activated protein kinase) and SIRT1 (Sirtuin 1) signaling pathway. Moreover, naturally occurring polyphenolic compounds found in various plants, including grapes, berries, and peanuts have also been shown to reduce OxS in the liver. For example, Resveratrol and Quercetin were shown to be combinatorially effective at reducing fatty liver, in a recent study investigating metabolic syndrome (MS) in rats (35). The mechanism was attributed to the over-expression of the master factor NrF2, which in turn led to the increase of antioxidant enzymes (catalase, peroxidases, glutathione-S-transferase, glutathione reductase) and GSH (reduced glutathione) recycling. Line 163

  1. What is authors' opinion about the utility of ASOs in the treatment monogenic diseases and multifactorial diseases. Namely, for what types of diseases, ASOs seem to be more effective based on results of studies described in Chapter 5 and 6. Please comment.

We appreciate this point and added text to line 460, “For the moment, ASO therapies may better suited for monogenic diseases, where a mutation in a single gene is responsible for the disease phenotype. Correcting or modulating the expression of an mRNA transcript may be a more parsimonious approach, leading to selective inhibition of the abnormal gene product. By contrast, cardiovascular diseases, diabetes, and many types of cancer are examples of multifactorial diseases where the challenge of controlling for multiple genetic variants, environmental factors, and complex biological pathways may pose undue technical challenges.”

Detail comments

  1. Provide the full name of used abbreviations:  "PD-(L)1, MAPT" in Table 1 

We have added the names as recommended

  1. In line 142  shortcut "8-OHG"  was applied, please provide its full name.

We have added the name as recommended, line 161